# Level-*k* Models and Overspending in Contests

**Malin Arve [1] and Marco Serena [2],*** 

[1]  Department of Business and Management Science, Norwegian School of Economics, 5045 Bergen, Norway; malin.arve@nhh.no
[2]  Department of Public Economics, Max Planck Institute for Tax Law and Public Finance, Marstallplatz 1, D-80539 München, Germany
*  Correspondence: marco.serena@tax.mpg.de

**Abstract:** The experimental evidence on contests often reports overspending of contest participants compared to the theoretical Nash equilibrium outcome. We show that a standard level-*k* model may rationalize overspending in contests. This result complements the existing literature on overspending in contests, and it bridges an open gap between the contest and auction literature. In fact, the literature on auctions often runs parallel to that on contests. Overbidding in auctions has also been documented empirically, and it has been shown that, in private-value auctions, such overbidding can be rationalized by level-*k* reasoning. We bridge the existing gap between the auction and contest literature by showing that overbidding may also be true in a theoretical contest environment with level-*k* reasoning.

**Keywords:** Tullock contests; level-*k* reasoning; overspending

**JEL Classification:** C72; D72

## 1. Introduction

The experimental literature on contests often reports significant overspending of subjects compared to the Nash equilibrium—see, [1–4], among others, and [5] for a survey of experimental research on contests. We propose a theoretical rationale for overspending in Tullock contests ([6]) which is based on the standard level-*k* model.

We adopt a standard level-*k* model, which works as follows. A player of depth of reasoning *k*, or a level-*k* player (henceforth, in short, "an *Lk*-player"), believes that the other players are $Lk − 1$, and thus "best responds" to this belief. In particular, *L1*-players best respond to all other players playing the strategy of *L0*-players, *L2*-players best respond to all other players playing the strategy of *L1*-players, and so on. (Some alternatives of behavior of types *L2* and higher have been proposed. For instance, in [7,8] while *L1*-players best respond to uniformly playing *L0*-players, *L2*-players best respond to a mixture of *L0*-players and *L1*-players, and so on. Another alternative is [9], where *Lk*-players best respond to an estimated mixture of players of lower *k*, via a one-parameter Poisson distribution. In the present paper, we choose the most simple and common approach that *Lk* best responds to all others being $Lk − 1$).

*L0*-players are nonstrategic in that they form no beliefs over the opponents' actions and rather play an instinctive reaction to the game. It is common in the literature to assume that *L0*-players uniformly mix over all possible actions ([7–14]) and this is backed up by empirical findings. An alternative approach is that *L0*-players choose a salient or intuitive action, also known as a "focal point". However, as reported by [13], "the evidence [. . . ] generally supports level-*k* models in which players anchor beliefs in a uniform random *L0*".

Furthermore, the literature on auctions often runs parallel to that on contests. Overbidding in auctions has also been documented empirically/experimentally (see, [15] for an overview) and can in private-value auctions be rationalized by level-*k* models ([12]).

We bridge the existing gap between the auction and contest literature by showing that overbidding can also be true in a contest environment due to level-*k* reasoning.

In general, there are two main branches of theoretical models that have been proposed to explain overspending. The first branch of models drops the standard specification of the expected utility function of subjects, e.g., assuming an extra non-monetary utility derived either from winning (see [16] for the case of contests and [17,18] for the case of auctions) or from relative payoff (e.g., [19]), or assuming that subjects assign a distorted value to the probability of winning (e.g., [20]).

The second branch of models to which this paper belongs suggests that players do not work out all the necessary steps to compute the Nash equilibrium, i.e., subjects are boundedly rational. Two approaches to bounded rationality models are common in the literature. The first is through quantal response equilibrium (QRE) models (see [21]), where the probability of a certain action is increased with the expected payoff of that action. Applications of QRE models to contests can be found in [22–24]. For auctions, overspending based on QRE is shown by [25]. The second is through level-*k* models; for auctions, [12] show that level-*k* models can rationalize overbidding. We provide the missing link that shows that a similar result can be obtained in the realm of contests.

To our knowledge, the only paper applying level-*k* models to contests is [26]. He studied a two-player symmetric Tullock contest and showed that overspending compared to the Nash equilibrium did not occur. This creates a divide between the auction and contest literature where the results are not the same from one literature to the other. We argue that [26] is a special case and that extending the framework to either more players or allowing for asymmetric players may yield an overspending result in line with the aforementioned parallelism between the literature on auctions and contests. A summary of the link between the contest and auction literature that we contribute to is summarized in Table 1.

**Table 1.** Does bounded rationality rationalize overspending. . .

|  | **QRE** | **Level-*k*** |
| --- | --- | --- |
| . . . in Auctions? | Yes: [25] | Yes: [12] |
| . . . in Contests? | Yes: [22–24] | No: [26] <br> Yes: present paper |

Our main argument relies on the observation that, in a two-player symmetric Tullock contest, the Nash equilibrium is situated at the peak of both players' best response functions. In such a situation, there is no scope for overspending since the Nash equilibrium is already at the maximum of the players' best response functions, and hence there is no belief for a best-responding player about the rival's behavior that would make them exert more effort than their Nash equilibrium one.

In turn, there is no assumption on the behavior of $L0$-players that would make an $Lk \geq 1$ player exert strictly more effort than in the Nash equilibrium, even if one were to cherry-pick the actions of the $L0$-players. As noted by [27] among others, one could always find an arbitrarily cherry-picked behavior of $L0$-players so as to rationalize any individually rational action of $L1$-players, if no further restrictions are imposed. However, in the present paper, we tie our hands by imposing that $L0$-players randomize uniformly over the space of positive efforts with a given upper-bound. Thus, our exercise of rationalizing overspending for any given such upper-bound is non-trivial.

However, we show that it is sufficient to drop one of the main two assumptions in [26]—namely, symmetry or having two players—to find overspending already for $L1$-players; the intuition relies on the Nash equilibrium no longer being at the peak of the best response function. In particular, we find that overspending occurs in two-player, sufficiently asymmetric contests and in three-player symmetric contests; see, respectively,

Propositions 2 and 3. Note that a minimal level of asymmetry is required; otherwise, the Nash equilibrium would be too close to the peak of the best response function.

Hence, despite the level-*k* model failing to provide a unifying explanation for the phenomenon of overspending in contests, we show that the level-*k* model can rationalize overspending when departing from the two-player symmetric setting. Therefore, we see our results as a step toward a robust hybrid model of player thinking, including other behavioral models, which could well explain overbidding across different contest settings.

Our theoretical results are also consistent with empirical evidence other than the possibility of overspending. First, the fact that overspending increases with the number of bidders (e.g., [28]) is consistent with our result that moving from two to three players makes overspending strictly positive for some parameter values. Second, the heterogeneity of contestants' behavior facing identical contests (e.g., [4]) is consistent with the fact that the depth of reasoning and the realization of the $L0$'s random behavior may differ across players.

The structure of the paper is as follows. In Section 2, we introduce the standard model of Tullock contest. In Section 3, we provide benchmark results for the Nash equilibrium as well as level-*k* results in the case of a two-player symmetric contest—thus, paralleling the results of [26]. In Section 4, we show that overspending may occur in two-player asymmetric contests, and in Section 5, in three-player symmetric contests. Section 6 discusses the results and briefly mentions the level-2 case. Proofs that do not follow directly from the main text are relegated to Appendix A.

## 2. Model

Consider a complete information Tullock-contest with $n$ risk-neutral players indexed by $i \in \{1, \ldots, n\}$. Players compete for a prize whose value, without loss of generality, normalizes to 1. Each player $i$ chooses effort level $e_i$ and has a probability of winning the prize equal to $p_i(e_1, \ldots, e_n) = e_i / E$, where $E = \sum_{j=1}^{n} e_j$ and $e_i \geq 0 \; \forall i \in \{1, \ldots, n\}$. If all efforts are 0, the prize is awarded with a fixed probability strictly less than 1 to each player. (This situation is never reached in equilibrium.) The cost of effort is linear, and the marginal cost equals $c_i > 0$ for player $i$. Hence, player $i$ chooses $e_i$ to maximize

$$\frac{e_i}{\sum_{j=1}^{n} e_j} - c_i e_i. \tag{1}$$

Throughout the paper, we denote, by $e^{NE}$, the Nash equilibrium of such a game, and by $e^{Li}$, the effort of $Li$-players in the level-*k* version of such a game. In level-*k* models, the most frequently observed types are typically $L1$ and $L2$, and higher levels are seldom observed. (For estimations of the distribution of subjects' levels of reasoning see for instance [11,29,30].) Since our ultimate goal is to test whether, at some $k$, overspending is rationalizable, and we find that at $k = 1$, a player already overspends, we only characterize the $L0$ and $L1$ here, and we briefly discuss the evolution of behavior of higher types in the final discussion.

While, in games with compact action space, the support over which $L0$-players randomly choose their action is unambiguously defined, in contests the strategy space is, potentially, the whole positive real line. Thus, we assume that $L0$-players randomize uniformly over the interval $[0, \bar{e}]$ with $\bar{e} > 0$. Note that we take an agnostic view on $\bar{e}$, which we allow to be lower, but also greater, than the valuation of the prize. In fact, some experimental evidence shows that subjects may bid more than the prize they might win, e.g., [2,4,31,32]. $\bar{e}$ can be interpreted as the initial endowment of money that subjects are often given in experiments, and the fact that $L0$-players choose randomly between 0 and their endowment is consistent with the empirical finding that overspending increases with the endowment (see the meta-study by [28]).

### 3. Benchmark: Nash Equilibrium and Two-Player Symmetric Contests with Level-*k* Reasoning

Before presenting our results, we present the main benchmark, the Nash equilibrium with fully rational players, as well as a generalization of the result for two-player symmetric contests with level-*k* reasoning in [26]. These are the two outcomes that we contrast our results with.

**Nash equilibrium under full rationality.** With two possibly heterogeneous players, the well-known unique Nash equilibrium of the full rationality model for the two players is

$$e_1^{NE} = \frac{a}{(1+a)^2 c_1} \text{ and } e_2^{NE} = \frac{1}{(1+a)^2 c_1}, \tag{2}$$

where we define $a = c_2/c_1$, which is thus the asymmetry between contestants. With $n$ homogeneous players with cost parameter $c_i = c$, $\forall i \in \{1, \dots, n\}$, the unique Nash equilibrium of the full rationality model is

$$e^{NE} = \frac{n-1}{n^2 c}. \tag{3}$$

A full characterization of the equilibrium with $n$ players and asymmetry is, for instance, in [33]. Even with just three asymmetric players, there are parameter constellations for which not all contestants exert positive effort, and these endogenous participation issues would confound the results of our paper. In fact, our ultimate goal is to verify the rationalizability of overspending in Tullock contests, and the special cases of three symmetric players and two asymmetric players suffice for this goal.

**Two-player symmetric contests with level-k reasoning.** We now consider the symmetric two-player contest with level-*k* reasoning and prove the impossibility of overspending result. This benchmark thus generalizes the impossibility found in [26] to any $(c, \bar{e})$, where $c$ is the marginal cost of effort for a player, as he studied the special case of $c = \bar{e} = 1$.

An $L1$-player believes that their rival is $L0$ and thus plays uniformly over $[0, \bar{e}]$. Therefore, the first-order condition (FOC) of $L1$-players of (1), when they believe that their rival's effort is uniformly distributed over $[0, \bar{e}]$, reads

$$\int_0^{\bar{e}} \frac{z}{\bar{e}(e^B + z)^2} dz = c$$

$$\iff \left. \frac{e^B + (z + e^B) \ln z + e^B}{z + e^B} \right|_{z=0}^{z=\bar{e}} = c\bar{e}$$

$$\iff \frac{e^B}{e^B + \bar{e}} + \log[e^B + \bar{e}] - 1 - \log[e^B] = c\bar{e}$$

$$\iff \log\left[\frac{e^B + \bar{e}}{e^B}\right] - \frac{\bar{e}}{e^B + \bar{e}} = c\bar{e}, \tag{4}$$

where $e^B$ denotes the equilibrium strategy of $L1$ players in the generalized Bernard model. Note that the left-hand side (LHS) of (4) is strictly decreasing in $e^B$, tends to $\infty$ as $e^B \to 0$, and tends to 0 as $e^B \to \infty$. Thus, a unique interior equilibrium $e^B$ exists and solves (4).

Since the LHS of (4) decreases in $e^B$, we obtain

$$
\begin{aligned}
e^B \quad &> \quad e^{NE} \\
&\Longleftrightarrow \quad \log\left[\frac{e^{NE} + \bar{e}}{e^{NE}}\right] - \frac{\bar{e}}{e^{NE} + \bar{e}} > c\bar{e}, \\
&\Longleftrightarrow \quad \log[1 + 4c\bar{e}] - \frac{4c\bar{e}}{1 + 4c\bar{e}} > c\bar{e}, \\
&\Longleftrightarrow \quad \log[1 + 4c\bar{e}] - \frac{5c\bar{e} + 4(c\bar{e})^2}{1 + 4c\bar{e}} > 0,
\end{aligned}
\tag{5}
$$

where the third line follows by the definition of $e^{NE}$ in (3) for the $n = 2$ case.

[26] assumes that $c = \bar{e} = 1$, for which the above condition is violated, yielding their result of impossibility of overspending. Routine algebra shows that the LHS of (5) decreases in $c\bar{e}$, and its value is 0 when $c\bar{e} = 0$; thus, (5) never holds for any $c, \bar{e} > 0$. Hence, we have proven the following generalization of impossibility of overspending in a symmetric two-player contest from [26].

**Proposition 1.** *Consider a symmetric Tullock contest with $n = 2$. Then, there is no pair $(c, \bar{e})$ such that the L1 level of effort is greater than the effort level at the fully rational Nash equilibrium. That is, $e^B \leq e^{NE}$.*

This impossibility result is a direct consequence of the fact that in a two-player symmetric contest the fully rational Nash equilbrium is at the peak of the players' best response functions. Thus, overspending as opposed to the effort level at the fully rational Nash equilibrium is impossible in such a setting, as already proven by [26] in the special case of $c = \bar{e} = 1$.

However, this result relies on two important assumptions, namely $n = 2$ and $c_1 = c_2 = c$. In the next two sections, we show that dropping either of these two assumptions may yield overspending relative to the Nash equilibrium.

## 4. Overspending in Two-Players Asymmetric Contest

The main point of this section is that introducing asymmetry in players' costs of effort rationalizes overspending. Without loss of generality, we focus on player 1 and thus consider an $L1$-player having marginal cost $c_1$.

An $L1$-player believes that their rival is $L0$ playing uniformly over $[0, \bar{e}]$. Therefore, it is straightforward to see that $c_2$ does not enter into the condition of optimal $e^{L1}$, and the FOC of $L1$-players writes

$$
\begin{aligned}
&\int_0^{\bar{e}} \frac{z}{\bar{e}(e^{L1} + z)^2} dz = c_1 \\
&\Longleftrightarrow \quad \frac{e^{L1}}{e^{L1} + \bar{e}} + \log[e^{L1} + \bar{e}] - 1 - \log[e^{L1}] = c_1\bar{e} \\
&\Longleftrightarrow \quad \log\left[\frac{e^{L1} + \bar{e}}{e^{L1}}\right] - \frac{\bar{e}}{e^{L1} + \bar{e}} = c_1\bar{e}.
\end{aligned}
\tag{6}
$$

As in the benchmark case with symmetric costs (note the similarity between (6) and (4)), a unique interior equilibrium $e^{L1}$ exists and solves (6).

Since the LHS of (6) decreases in $e^{L1}$, using the definition of $e_1^{NE}$ in (2), we obtain

$$
\begin{aligned}
e^{L1} \quad &> \quad e^{NE} \\
\iff \quad &\log\left[\frac{e^{NE} + \bar{e}}{e^{NE}}\right] - \frac{\bar{e}}{e^{NE} + \bar{e}} > c_1\bar{e} \\
\iff \quad &\log\left[1 + \frac{(a+1)^2 c_1\bar{e}}{a}\right] - \frac{(a+1)^2 c_1\bar{e}}{a + (a+1)^2 c_1\bar{e}} > c_1\bar{e}.
\end{aligned}
\tag{7}
$$

We plot, in Figure 1, condition (7) in two dimensions, namely $(c_1\bar{e}, a)$. The shaded area represents combinations $(c_1\bar{e}, a)$ such that the $L1$ level of effort is greater than the Nash equilibrium level.

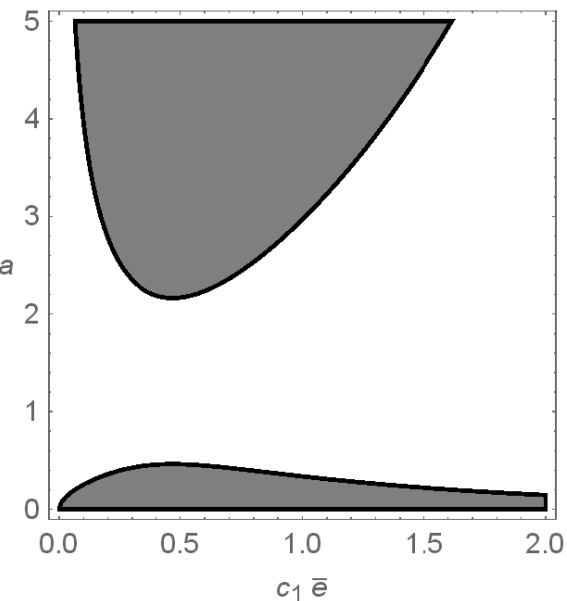

**Figure 1.** Region in the $(c_1\bar{e}, a)$−space where in a two-player contest $e^{L1} > e^{NE}$—that is, the $L1$-level of effort is greater than the Nash equilibrium level.

If $a \to \infty$, the LHS of (7) tends to $\infty$, and thus for any finite value of $c_1\bar{e}$, $\exists a$ sufficiently large, (7) holds and overspending occurs. Proving the opposite, namely that for any finite value of $a \geq \bar{a}$, $\exists c_1\bar{e}$ makes (7) hold, is more subtle, and we prove it in Lemma A1 in the Appendix A, which also characterizes $\bar{a}$ (i.e., $\bar{a} \approx 2.16258$). As $a$ measures the ratio between $c_2$ and $c_1$ (recall that $a = c_2/c_1$), by symmetry, we also find a lower bound $\underline{a}$ below which this is true. Thus, we have proven the following.

**Proposition 2.** *Consider a Tullock contest with $n = 2$. Out of the three parameters $(c_1, c_2, \bar{e})$, fixing the value of two of these parameters, there exists a value of the remaining parameter such that the $L1$ level of effort is greater than the Nash equilibrium level, i.e., $e^{L1} > e^{NE}$. If $c_1$ and $c_2$ are the chosen parameters, it also has to be that $c_2/c_1 \notin (\underline{a}, \bar{a})$ with $\underline{a} \approx 0.4624$ and $\bar{a} \approx 2.1626$.*

Note that a minimum level of asymmetry between players is needed to rationalize overspending because, if instead costs were close enough (the white region of Figure 1), the Nash equilibrium would be too close to the peak of the best response function, and thus we would not be able to obtain the overspending result. Overspending in the asymmetric level-$k$ Tullock contest can happen both by a weak or strong player who has sufficiently low or high cost compared to their competitor.

One remark is in order. Within the family of two-players asymmetric contests, we provided a characterization for when *an* $L1$-player overspends as opposed to the Nash equilibrium. It is not clear whether it could be that, if both players are $L1$, they both

overspend. In other words, fixing for simplicity $\bar{e} = 1$, is there a pair $(c_1, c_2)$ such that, if both players 1 and 2 were $L1$, they both overspend? This question boils down to verifying whether condition (7) and its mirror image, where $c_1$ and $c_2$ (which enters through $a$) are swapped can simultaneously hold. The answer is positive, and an example is $(c_1, c_2) = (1, 1/10)$ as one can readily verify.

## 5. Overspending in Three-Players Symmetric Contest

The main point of this section is that introducing a third player in a symmetric contest rationalizes overspending. An $L1$-player believes that their two rivals are $L0$-players mixing uniformly over $[0, \bar{e}]$. A Tullock contest is an aggregative game, as pioneered by [34], in that (1) depends only on $e_1$ and on the sum of the rival's efforts $e_2 + e_3$. For this reason, $L1$-players who are up against two rivals behave as if up against one rival whose effort follows a triangular distribution $Z$ in $[0, 2\bar{e}]$—that is, the distribution of the sum of the two uniform random variables both in $[0, \bar{e}]$, namely:

$$f_Z(z) = \begin{cases} \frac{z}{\bar{e}^2} & \text{if } z \in [0, \bar{e}], \\ \frac{2\bar{e} - z}{\bar{e}^2} & \text{if } z \in [\bar{e}, 2\bar{e}]. \end{cases} \tag{8}$$

Therefore, the FOC of the $L1$-players writes

$$\int_0^{\bar{e}} \frac{z^2}{\bar{e}^2 (e^{L1} + z)^2} dz + \int_{\bar{e}}^{2\bar{e}} \frac{(2\bar{e} - z)z}{\bar{e}^2 (e^{L1} + z)^2} dz = c, \tag{9}$$

where the first integral accounts for the part of (8) where $z \in [0, \bar{e}]$ and the second integral accounts for the part of (8) where $z \in [\bar{e}, 2\bar{e}]$. Routine algebra relegated to Lemma A2 in the Appendix A shows that (9) is equivalent to

$$e^{L1} \log[e^{L1}] - (\bar{e} + 2e^{L1}) \log[\bar{e} + e^{L1}] + (\bar{e} + e^{L1}) \log[2\bar{e} + e^{L1}] = \frac{\bar{e}^2 c}{2}. \tag{10}$$

As we prove in Lemma A3 in the Appendix A, the LHS of (10) decreases in $e^{L1}$. Second, as $e^{L1} \to 0$, the LHS of (10) approaches $\bar{e} \log 2 > 0$, and as $e^{L1} \to \infty$, it approaches 0. Thus, there exists a unique equilibrium. Such an equilibrium is interior by $2 \log 2 > \bar{e}c$.

Since the LHS of (10) decreases in $e^{L1}$, $e^{L1} > e^{NE}$ if and only if the LHS of (10) with $e^{NE}$ instead of $e^{L1}$ is greater than the RHS.

We next show that $\forall \bar{e} > 0$, there exists a $c > 0$ such that $e^{L1} > e^{NE}$, and vice versa, $\forall c > 0$ there is an $\bar{e} > 0$ such that $e^{L1} > e^{NE}$. An easy way to do so is to consider

$$\bar{e}c = \frac{1}{4}, \tag{11}$$

in condition (10) above since for any $\bar{e}$ or $c$ this pins down the value of the other parameter. Note that (11) satisfies the condition for an interior solution $2 \log 2 > \bar{e}c$ found above. The intuition behind condition (11) is that in order to rationalize overspending one has to trade-off $\bar{e}$ and $c$. In fact, high expected effort from $L0$-players (i.e., high $\bar{e}$) discourages $L1$-players and thus has to be compensated by low marginal cost of effort $c$ to guarantee that she still exerts high effort. Similarly, costly effort (i.e., high $c$) discourages $L1$-players; however, this is compensated when $L0$-players exert sufficiently low effort (i.e., low $\bar{e}$).

As stated above, $e^{L1} > e^{NE}$, if and only if the LHS of (10) with $e^{NE}$ instead of $e^{L1}$ is greater than the RHS, and thus, under condition (11) and using the definition of $e^{NE}$ in (3) for the special case of $n = 3$, we obtain that

$$
\begin{aligned}
e^{L1} \quad &> \quad e^{NE} \\
\iff \quad & \frac{8\bar{e}}{9} \log\left[\frac{8\bar{e}}{9}\right] - (\bar{e} + \frac{16\bar{e}}{9}) \log\left[\bar{e} + \frac{8\bar{e}}{9}\right] + (\bar{e} + \frac{8\bar{e}}{9}) \log\left[2\bar{e} + \frac{8\bar{e}}{9}\right] > \frac{\bar{e}}{8} \\
\iff \quad & \frac{8}{9} \log\left[\frac{8\bar{e}}{9}\right] - \frac{25}{9} \log\left[\frac{17\bar{e}}{9}\right] + \frac{17}{9} \log\left[\frac{26\bar{e}}{9}\right] > \frac{1}{8} \\
\iff \quad & \frac{8}{9} \log\left[\frac{8\bar{e}}{9}\right] - \frac{8}{9} \log\left[\frac{17\bar{e}}{9}\right] - \frac{17}{9} \log\left[\frac{17\bar{e}}{9}\right] + \frac{17}{9} \log\left[\frac{26\bar{e}}{9}\right] > \frac{1}{8} \\
\iff \quad & \frac{8}{9} \log\left[\frac{8}{17}\right] + \frac{17}{9} \log\left[\frac{26}{17}\right] > \frac{1}{8},
\end{aligned}
$$

which holds true. Thus, we have proven the following:

**Proposition 3.** *Consider a symmetric Tullock contest with $n = 3$. Out of the two parameters $(c, \bar{e})$, fixing the value of one of these parameters, there exists a value of the remaining parameter such that the L1 level of effort is greater than the Nash Equilibrium level, i.e., $e^{L1} > e^{NE}$.*

Note that having three players as opposed to two players shifts the Nash equilibrium sufficiently far from the peak of the best response function, and this may yield per se sufficient asymmetry to rationalize overspending. In other words, we do not need further assumptions on sufficient asymmetry, such as $a \geq \bar{a}$ or $a \leq \underline{a}$ in Proposition 2.

In Figure 2, we plot in the $(c, \bar{e})$-space the region where $e^{L1} > e^{NE}$. The dashed line is condition (11), which we used to prove Proposition 3.

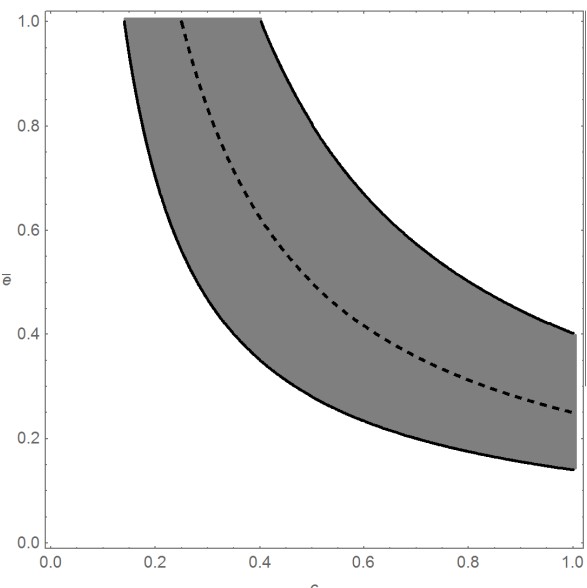

**Figure 2.** Region in the $(c, \bar{e})$−space where in a three-player contest $e^{L1} > e^{NE}$—that is, the L1 level of effort is greater than the Nash equilibrium level. The dashed line is condition $\bar{e}c = \frac{1}{4}$.

## 6. Discussion and *L2*

Overbidding in auctions as opposed to the Nash equilibrium has been documented empirically/experimentally and can, in private-value auctions, be rationalized by level-*k* models ([12]). Overspending in contests as opposed to the Nash equilibrium has also been documented empirically/experimentally. We show that level-*k* models can rationalize such overspending in Tullock contests. We thus bridge the existing gap between the auction

and contest literature, two types of literature that usually work in parallel, by showing that overbidding can also be true in a contest environment due to level-$k$ reasoning.

The main analysis conducted in the present paper focused on the $L1$ level of effort since this was found to be sufficient to rationalize overspending. However, one might wonder about the level of effort of $Lk$-players with $k > 1$. In particular, since we know from empirical evidence that the most frequently observed types are typically $L1$ and $L2$, it is important to focus also on $k = 2$, as we do in the remainder of the paper.

An $L2$ player believes that their opponent is an $L1$ player, and thus "best responds" to this belief. Such a best response, which maps the effort of the $L1$ into the effort of the $L2$, is a strictly concave function with a unique maximum $e^{\max}$. More precisely, the best response function of $Lk$-players to a certain effort of $Lk-1$-players is

$$e_1^{Lk} = BR_1\left(e_2^{Lk-1}\right) \equiv -(n-1)e_2^{Lk-1} + \sqrt{\frac{(n-1)e_2^{Lk-1}}{c_1}}. \tag{12}$$

In fact, this characterizes the behavior of any level-$k$ type. Note that $BR_1(\cdot)$ reaches its maximum when $e_2^{Lk-1} = \frac{1}{4c_1(n-1)} \equiv e_2^{\max}$. Overspending remains possible at $L2$ in both cases of interest to our analysis, as we explain in what follows, and it may also be possible that both the $L1$- and $L2$- players overspend.

First, if there are *two asymmetric* players, calling 2 the $L1$-player and 1 the $L2$-player, then the condition for overspending of the $L1$-player is $e_2^{L1} > c_1/(c_1+c_2)^2 = e_2^{NE}$, and the condition for overspending of the $L2$-player is $e_1^{L2} = BR_1\left(e_2^{L1}\right) > c_2/(c_1+c_2)^2 = e_1^{NE}$. Using the definition of $BR_1(\cdot)$ in (12), we can rewrite this last condition as

$$e_2^{L1} \quad \in \quad \left[\frac{c_1}{(c_1+c_2)^2}, \frac{c_2^2}{c_1(c_1+c_2)^2}\right] \text{ if } c_1 < c_2, \tag{13}$$

$$e_2^{L1} \quad \in \quad \left[\frac{c_2^2}{c_1(c_1+c_2)^2}, \frac{c_1}{(c_1+c_2)^2}\right] \text{ if } c_1 > c_2. \tag{14}$$

It is clear that, if we want to rationalize both $L1$- and $L2$-players' overspending, the condition for overspending of the $L1$-player ($e_2^{L1} > e_2^{NE}$) is compatible only with the $c_1 < c_2$ case of the above condition—that is, (13). In other words, it is possible that both $L1$- and $L2$-players overspend only if the $L1$-player is weaker than the $L2$-player. Conversely, if instead the $L1$-player is stronger than the $L2$-player ($c_1 > c_2$), then the overspending condition of the $L2$-player (14) implies that the $L1$-player underspends ($e_2^{L1} < e_2^{NE}$).

Second, if there are *three symmetric* players, dropping the player-specific subscripts from the notation, we obtain that $e^{NE} = 2/(9c)$ and $e^{\max} = 1/(8c)$, and therefore $BR(e^{\max}) = 1/(4c)$, and, similarly to the two asymmetric player case, $e^{NE} \leq BR(e^{\max})$—an $L2$ overspends. It is also interesting to know how $e^{Lk}$ with $k \geq 2$ evolves if $e^{L1} > e^{NE}$. Normalizing marginal costs to 1, one can show that

$$BR\left(e^{Lk-1}\right) \quad \geq \quad e^{NE} \iff e^{Lk-1} \in \left[\frac{1}{n^2(n-1)}, \frac{n-1}{n^2}\right],$$

$$e^{Lk-1} \quad \geq \quad e^{NE} \iff e^{Lk-1} \geq \frac{n-1}{n^2}.$$

Hence, $e^{Lk-1} \geq e^{NE}$ implies $e^{Lk} \leq e^{NE}$, but that the converse is not necessarily true; if $e^{Lk-1} \leq e^{NE}$, then we could obtain $e^{Lk} \leq e^{NE}$ or $e^{Lk} \geq e^{NE}$. For instance, consider the $n = 3$ case. If $e^{Lk-1} \geq e^{NE} = 2/9$, then $e^{Lk} \leq e^{NE}$. If instead $e^{Lk-1} \leq e^{NE}$, then $e^{Lk} \gtrless e^{NE} \iff e^{Lk-1} \lessgtr 1/\left(n^2(n-1)\right)$. Hence, a general analysis of the evolution of $Lk$ is found not to be characterized by simple rules (such as alternations of overspending and underspending) and, importantly, beyond the scope of the paper, which is to verify the rationalizability of overspending in Tullock contests. Extending the two-symmetric-

player setup of [26] to the cases of three symmetric and two asymmetric $L1$-players proved sufficient for this goal.

**Author Contributions:** Conceptualization, M.A. and M.S.; methodology, M.A. and M.S.; software, M.A. and M.S.; validation, M.A. and M.S.; formal analysis, M.A. and M.S.; investigation, M.A. and M.S.; resources, M.A. and M.S.; writing—original draft preparation, M.A. and M.S.; writing—review and editing, M.A. and M.S.; visualization, M.A. and M.S.; project administration, M.A. and M.S. All authors have read and agreed to the published version of the manuscript.

**Funding:** This research received no external funding.

**Institutional Review Board Statement:** Not applicable.

**Informed Consent Statement:** Not applicable.

**Data Availability Statement:** Not applicable.

**Acknowledgments:** We are grateful to Cédric Wasser and to the participants of the fifth informal research meeting at the Max Planck Institute for Tax Law and Public Finance for useful comments. Errors are ours.

**Conflicts of Interest:** The authors declare no conflict of interest.

## Appendix A

**Lemma A1.** $\forall a \geq \bar{a}$ and $\forall a \leq \underline{a}$, $\exists c_1 \bar{e}$ such that (7) holds. In particular, $\bar{a} \approx 2.1626$ and $\underline{a} = 0.4624$.

**Proof of Lemma A1.** Consider Figure 1. We characterize the unique minimum $\bar{a}$ of the function in the LHS of (7). The minimum is where

$$
\begin{aligned}
0 \quad &= \quad \frac{\partial}{\partial c_1 \bar{e}} \left( \log \left( 1 + \frac{(a+1)^2 c_1 \bar{e}}{a} \right) - \frac{(a+1)^2 c_1 \bar{e}}{a + (a+1)^2 c_1 \bar{e}} - c_1 \bar{e} \right) \\
\iff \quad 0 &= \frac{a(a+1)^2}{a + (a+1)^2 c_1 \bar{e}} - \frac{(a+1)^2}{\left( a + (a+1)^2 c_1 \bar{e} \right)^2} - 1 \\
\iff \quad 0 &= \left( a + (a+1)^2 c_1 \bar{e} \right)^2 - a(a+1)^2 \left( a + (a+1)^2 c_1 \bar{e} \right) + (a+1)^2
\end{aligned}
$$

Furthermore, the solution of the above together with (7) characterizes the unique minimum. Namely, $(c_1 e, \bar{a}) \approx (0.467586, 2.1626)$. Swapping players' indexes, one can find the maximum of the bottom shaded area of Figure 1, $\underline{a} = 1/\bar{a} = 0.4624$. □

**Lemma A2.** *Equation (9) is equivalent to (10).*

**Proof of Lemma A2.** Note that (9) can be written as

$$
\int_0^{\bar{e}} \left( 1 - \frac{2e^{L1}z + \left(e^{L1}\right)^2}{\left(e^{L1} + z\right)^2} \right) dz - \int_{\bar{e}}^{2\bar{e}} \left( 1 - \frac{2e^{L1}z + \left(e^{L1}\right)^2}{\left(e^{L1} + z\right)^2} \right) dz + \int_{\bar{e}}^{2\bar{e}} \frac{2\bar{e}z}{\left(e^{L1} + z\right)^2} dz = \bar{e}^2 c,
$$

or, calculating the values of the integrals,

$$
\begin{aligned}
-e^{L1} \left( \frac{-\bar{e}}{e^{L1} + \bar{e}} + 2\log \frac{e^{L1} + \bar{e}}{e^{L1}} \right) &+ e^{L1} \left( \frac{e^{L1}}{e^{L1} + 2\bar{e}} - \frac{e^{L1}}{e^{L1} + \bar{e}} + 2\log \frac{e^{L1} + 2\bar{e}}{e^{L1} + \bar{e}} \right) \\
&+ 2\bar{e} \frac{e^{L1}}{e^{L1} + 2\bar{e}} - \frac{e^{L1}}{e^{L1} + \bar{e}} + \log \frac{e^{L1} + 2\bar{e}}{e^{L1} + \bar{e}} = \bar{e}^2 c.
\end{aligned}
$$

Collecting the terms in the above expression, we obtain

$$e^{L1}\left(\frac{2\bar{e}^2}{(e^{L1}+2\bar{e})(e^{L1}+\bar{e})}+2\log\frac{e^{L1}+2\bar{e}}{e^{L1}+\bar{e}}-2\log\frac{e^{L1}+\bar{e}}{e^{L1}}\right)$$
$$+2\bar{e}\left(-\frac{e^{L1}\bar{e}}{(e^{L1}+2\bar{e})(e^{L1}+\bar{e})}+\log\frac{e^{L1}+2\bar{e}}{e^{L1}+\bar{e}}\right)=\bar{e}^2c$$

or

$$e^{L1}\left(\log\frac{e^{L1}+2\bar{e}}{e^{L1}+\bar{e}}-\log\frac{e^{L1}+\bar{e}}{e^{L1}}\right)+\bar{e}\left(\log\frac{e^{L1}+2\bar{e}}{e^{L1}+\bar{e}}\right)=\frac{\bar{e}^2c}{2},$$

which is equivalent to (10). $\quad\square$

**Lemma A3.** *The LHS of (10) decreases in $e^{L1}$.*

**Proof of Lemma A3.** The statement of the lemma is equivalent to

$$\frac{\partial}{\partial e^{L1}}\left(e^{L1}\log\left(e^{L1}\right)-(a+2e^{L1})\log\left(a+e^{L1}\right)+(a+e^{L1})\log\left(2a+e^{L1}\right)\right)<0$$

$$\iff \log\left(e^{L1}\right)+1-2\log\left(a+e^{L1}\right)-\frac{a+2e^{L1}}{a+e^{L1}}+\log\left(2a+e^{L1}\right)+\frac{a+e^{L1}}{2a+e^{L1}}<0$$

$$\iff \log\left(e^{L1}\right)+1-2\log\left(a+e^{L1}\right)-1+\frac{e^{L1}}{a+e^{L1}}+\log\left(2a+e^{L1}\right)+1-\frac{a}{2a+e^{L1}}<0$$

$$\iff 1+\frac{e^{L1}}{a+e^{L1}}-\frac{a}{2a+e^{L1}}+\log\left(\left(\frac{e^{L1}}{a+e^{L1}}\right)\left(\frac{2a+e^{L1}}{a+e^{L1}}\right)\right)<0$$

$$\iff \frac{a^2}{(a+e^{L1})(2a+e^{L1})}+\log\left(\left(1-\frac{\bar{e}}{a+e^{L1}}\right)\left(1+\frac{a}{a+e^{L1}}\right)\right)<0$$

$$\iff \frac{a^2}{(a+e^{L1})(2a+e^{L1})}+\log\left(1-\left(\frac{a}{a+e^{L1}}\right)^2\right)<0$$

for which it suffices, using the logarithm inequality $\log(1-x)\le -x$ with $x\in[0,1)$, that

$$\frac{a^2}{(a+e^{L1})(2a+e^{L1})}-\left(\frac{a}{a+e^{L1}}\right)^2<0,$$

which trivially holds. $\quad\square$

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
