# Peer review of "Level-k Models and Overspending in Contests"

_games, doi:10.3390/g13030045_

Round 1

Reviewer 1 Report

Referee report on: “Level-k models and overspending in contests”

The authors incorporate the k-level model into the Tullock contest. In particular, they consider a Tullock contest with two types of players - one randomizes over her strategy space (L0 type) and the other is rationale, i.e., plays her best response (L1 type). There are two main results: 1) with 2 symmetric players the equilibrium effort of the L1 type cannot exceeds the equilibrium effort of a player in the corresponding contest with two rationale players, and 2) with 2 sufficiently asymmetric players, or three symmetric players with sufficiently small marginal cost, the equilibrium effort of the L1 type exceeds the equilibrium effort of that player in the corresponding contest with rationale players. The intuition is nice and presented very clearly – the equilibrium effort in the two-player symmetric contest with rational players, is on the pick of the player’s best response curve, and therefore, in this special case it is impossible that the effort increases because of different levels of reasoning.

Overall evaluation: I enjoyed reading the paper. It is interesting and makes an important contribution to the literature. Previous studies show that the k-level models result in overspending in auctions but not in contests. By contrast, in this paper the authors show that, this is true with certainty only in the special case of the two-player symmetric Tullock contest.

 Some minor comments:

  1. In footnote 4, the word “recent” should be drop.
  2. In the introduction it is written: “The first branch of models drops the standard specification of the utility function of subjects..”

Instead of utility it should be expected utility, since Baharad and Nitzan (2008) distort probabilities.

  1. Page 7, first line, it should be sufficient.
  2. To help the reader, the integration in (4) can be done in more details. `

Author Response

Many thanks for your important comments, criticisms, and suggestions. As described in more detail below, we have taken them all into account with this revision. We believe addressing your comments significantly improved the quality of the paper. Please see the attachment for the point-by-point response.

Reviewer 2 Report

In this paper the authors apply level-k reasoning to explain overbidding in two player symmetric and asymmetric contests, as well as in three player symmetric contests. In this sense, the authors connect the streams of literature on level-k reasoning, auctions, and contests.

Overall, I think the authors overestimate the amount of information an audience can realistically appreciate and to what extent they are willing/capable to fill the gaps. Maybe a careful re-writing of the analyses which would allow the reader to better follow along would help convey the general message of the paper. In this sense, it is unfortunate that the interesting and relevant contribution that the paper sets out to make, gets lost in confusion about notation, algebra and a lack of a coherent storyline.

On a sidenote, it would have been convenient to have line numbers for reviewing the manuscript.

Specific comments, ordered as I come across in the paper, not in importance.

  1. Abstract: The abstract is very brief. Maybe the authors may want to open with the general relevance of the topic.
  2. Page 3, bottom paragraph: I don't quite get the message here. What do I have to imagine about the expected behaviour of L0? If an L0 player truly randomises, this would mean an L1 player optimises with respect to the midpoint of the endowment as spending level of the L0 player. Does this always lead to the symmetric Nash equilibrium as outcome? To me this is not trivial to see.
  3. Page 4, bottom paragraph: You present results for symmetric two-player contests, asymmetric three player contests, and symmetric three-player contests. This obviously begs the question: What about the rest? What about asymmetric three-player contests and what about n-player contests with n>3?
  4. Page 7, Equation (4): I find it not intuitive where this FOC comes from, particularly what z is. Would I be correct in assuming that this equation is the FOC for Equation (1) with some assumption about the L0-players' behaviour? If space restrictions apply, maybe consider presenting some intuition.
  5. Page 7, Equation (5), including text preceding it: Why is equation (3) equal to $c\bar{e}$ when n=2? If I plug n=2 into Equation (3), I get 1/4c.
  6. Page 8, after Proposition 1 ``overspending is impossible in such a setting’’: Do you mean overspending, i.e. spending more than the NE cannot be an equilibrium for level-k rational players? Because technically, I think overspending would be possible. This may be a semantic issue, but I did get confused about this already when coming across it in the Introduction.
  7. Page 9, Equation (7): Where does this a come from? What is the intuition of a? It seems that it is defined further down in the text, but it would need to be properly introduced.
  8. Page 10, top paragraph, ``because a… and $c_{2}$’’: Figure 1 only plots $a\geq0$. Does this mean that it measures the absolute relative distance?
  9. Page 10, equation at the bottom of the page:I don’t understand why the cases are necessary and what the intution of the functional forms are.
  10. Page 11, first sentence: To me, it is not trivial to see why (8) follows from the above. Especially, can you provide an intuition on the second part of Equation (8)?
  11. Page 12, first equation: Why is this $\frac{\bar{e}}{8}$? When applying n=3 to Equation (3), I get 2/9c.

Editorial comments, ordered as I come across in the paper, not in importance.

  1. Page 1, bottom paragraph: ``…short a Lk-player’. It should be ``an''. The use of a/an in English is oriented towards how the following word is pronounced. Also, upon first reading, it was not clear to me what the ``L’’ stands for here. Later it becomes clear that it abbreviates ``level’’. Maybe you can find an elegant way to introduce this notation efficiently.
  2. Page 3, bottom paragraph: best reply function is usually referred to as best response function. This appears again later in the text. Please change to the usual notation.
  3. Page 5: Sometimes you employ k as a ``normal’’ text letter, at other times you employ it as a variable, i.e. level-$k$ in Latex. It is probably nicer to always use the same way of notation, probably the variable-version.
  4. Page 8, bottom paragraph: It should be ``An L1-player’’.
  5. Page 13, bottom paragraph, ``that the her opponent’’: Please proofread

Author Response

(The authors gave the same response as above.)

Reviewer 3 Report

Please see the attached report.

Author Response

(The authors gave the same response as above.)

Round 2

Reviewer 2 Report

Thank you for considering my comments and questions. I feel that most of my concerns have been addressed. A few minor points of attention remain with respect to the same points as raised in the earlier review round.

1. Good

2. Good

3. You say that you clarify at the start of Section 5 that an n-player game would be beyond the scope of the paper, which is fine. If possible, you may consider conveying this information earlier in your paper.

4. You say ``We now better specify where (4) comes from''. I fail to see the change in the text that should serve to improve the specification of (4). Generally, providing a new manuscript with track changes would make my life as reviewer easier. Further, you mention that ``Indeed, z is the possible value of the effort of the L1-player's rival''. If I am not mistaken, this is not explicitly mentioned in the manuscript. Please provide this information and define z explicitly.

5. Okay

6. I believe we mean the same thing. I associate the term ``impossible'' more with being at the boundary, i.e. it would be technically impossible to spend more. For example, when a decision space is [0,1] it is impossible to spend more than 1, because it is beyond the decision space. In a symmetric Tullock contest with n=2 and c=1, there is an internal solution at e^{NE}=0.25. It is possible to spend more than that, it can just not be an equilibrium.

7. Thank you for clarifying this. Indeed now I see that a has been defined in a half-sentence three pages earlier. Have you considered providing a list of variables used? At times I find myself scrolling through the manusript searching for the definition of one of the many variables used in this study.

8. Indeed this confusion on my part has stemmed from having overlooked the definition of a in the section before this. Thank you for clarifying this.

9. I still do not understand where the functional form comes from, i.e. why specifically z/(e^2) to the left of the mode and (2\bar{e}-z) / \bar{e}^2 to the right of the mode. If this is clear to the editor and the other reviewers then I will yield and attribute this to my lack of knowledge on triangular distributions.

10. This is not clear to me. Are you trying to produce the CDF of (8) in equation (9)? If so, why is (9) the FOC of (8)? Should it not in fact be the other way around?

11. Okay
